# Willingness and Its Associated Factors for Blood Donation in Gondar Town, Northwest Ethiopia: A Community-Based Cross-Sectional Study

Agerie Mengistie Zeleke [1],* and Zelalem Nigussie Azene [2]

1   Department of Midwifery, School of Public Health, Teda Health Science College,
    Gondar P.O. Box 790, Ethiopia
2   Department of Women's and Family Health, School of Midwifery, College of Medicine and Health Sciences,
    University of Gondar, Gondar P.O. Box 196, Ethiopia
*   Correspondence: ageriemengistie@gmail.com

**Abstract:** Background: Although the World Health Organization recommends 100% willingness for blood donation, the percentage of blood collected from willing blood donors and the average annual blood collection rate is extremely low in Ethiopia. Adults can serve as an essential pool formeeting the demand of safe blood. Thus, this study is aimed at examining willingness and its associated factors for blood donation among the adult population in Gondar town, Northwest Ethiopia. Methods: A community-based cross-sectional study was conducted on548 respondents from 1–30 October2021. Multistage sampling techniques were used to select the study participants. The data were collected using an interviewer-administered structured questionnaire. Epi-data version 4.6 and SPSS version 23 software was used for data entry and analysis, respectively. A binary logistic regression (bivariable and multivariable) was performed to identify the statistically significant variables. Results: Less than half, 45.3% (95% CI: 41.4, 49.9), of the study participants hadthe willingness to donate blood. Participants who were renters of their own houses [AOR: 3.19; (95% CI: 2.09, 4.62)], had a history of blood donation practice [AOR: 1.90(95% CI: 1.16, 3.19)], had witnessed blood being donated [AOR: 2.56 (95% CI:1.65, 6.95)], had a history of relatives who have died through blood loss [AOR: 2.28 (95% CI:1.19, 4.36)], and had good knowledge [AOR: 2.23; (95% CI: 1.49, 3.34)] were more willing to donate blood, with these factors being the significant predictors. Conclusions: Generally, willingness towards blood donation is low in the study area. In order to increase community willingness to donate blood, healthcare providers, national blood banks, and transfusion agencies should design strategies to promote and motivate their communities. In addition to this, participants should receive information on the health benefits of donating blood, the volume of blood donated, and the number of patients benefiting from a single unit of blood donated.

**Keywords:** willingness; blood donation; community



## 1. Introduction

Blood is a life-sustaining fluid that transports nutrients and oxygen to the cells and moves metabolic wastes from the cells [1].A large volume of blood could be lost as a result of a variety of serious conditions, including obstetric hemorrhages, surgery, trauma, long-term chemotherapy, as well as anemia from medical or hematologic conditions [2,3]. Besides, children being treated for cancer, premature infants, and children having heart surgery need blood to survive [4]. Hence, blood transfusion from generous donors is an essential component of modern healthcare systems for saving millions of lives each year [5]. However, blood donation services are facing shortages, and the overall demand for blood is rising alarmingly throughout the world, particularly in low-income countries [4].

The World Health Organization recommends all the adult willingness to donate blood should focus on young people in order to achieve 100%, non remunerated voluntary blood

donation and should be self-sufficient in blood products with unpaid donors [6]. This is because young people are healthy, active, self-motivated, and open and constitute a greater proportion of the population [6].

Globally, an estimated 80 million units of blood were collected. Of this, only 38% was collected from developing countries [7]. In Sub-Saharan Africa, an estimated 18 million units of safe blood are required per year. Yet only 15% of the required blood was collected [8,9] and this is because, in Sub-Saharan Africa countries, blood donors usually donate when families or friends want a blood transfusion. However, in the most developed countries, most blood donors are voluntary, nonpaid donors who donate blood for their population [10].

Though Ethiopia has developed the Red Cross Society (ERCS) association for developing blood banking services, an adequate amount of blood supply has remained a challenge in Ethiopia. This indicates that, in Ethiopia, the 223,000 units of blood that were collected in 2019/20 only met 22% of its need, as per the standard set by the World Health Organization. The daily amount of approximately 1100 units of blood collected is a tremendous shortfall when compared to the 18,000 units of blood recommended [11]. This indicates that the availability of an adequate amount of blood is still a challenge for meeting the increased demand for long-term therapies and medical or hematological conditions, trauma, and chemotherapy in Ethiopia [12]. This problem may exist due to the increased number of victims who need blood. In order to successfully overcome this challenge, there is a need for evidence-based strategies and innovation concerning the awareness of the community towards blood donation [13].

Some previous studies among professionals revealed that the prevalence of a willingness to donate blood ranges from (58.1–78.1%)among health professionals(medical doctors, nurses, and midwives)in Ethiopia [14,15] and in Nigeria (59.3–73%) [16,17]. According to the study done in Jordan, poor knowledge and negative attitudes toward blood donation were barriers [18]. Beliefs, personal motivations, religion, residence, sex, and educational status were also predictors of a willingness to donate blood [7,19–21].

As far as we can tell, there has been much effort to increase the prevalence of a willingness to donate blood, including health education, public awareness creation in schools and streets, campaigns, and through different media (radio and TV) [22,23]. However, a willingness to donate blood is still the most common problem for the low proportion of blood donation practice in Ethiopia [24]. Therefore, the aim of this study was to assess the willingness to donate blood and its associated factors among adults in Gondar town, Northwest Ethiopia.

## 2. Materials and Methods

### 2.1. Study Design and Time Frame

A community-based cross-sectional study was conducted from 1–30 October 2021.

### 2.2. Study Setting

The study was conducted in Gondar town, Amhara Regional State in Northwest Ethiopia. Gondar town is located about 748 km Northwest of Addis Ababa (the capital of Ethiopia). It is divided into 12 administrative areas (sub cities), which consist of 21 kebeles (the smallest administrative units in Ethiopia). Ii is one of the most ancient and largely populated towns in the country. The town now has one comprehensive specialized hospital and eight health centers, and one or two general hospitals that provide health services to the population. According to the Ethiopian central statistics agency, the projected total population of the town in the year 2021 was 378,000 [25].

### 2.3. Participants

The source population were all adults aged 18–65 years residing in the town, while the study population were all adults aged 18–65 years residing in the town at least for six months during the data collection period.

Exclusion Criteria

Study participants who were critically ill, mentally ill, pregnant women, weighedless than 50 kg, and had transmittable diseases, like HIV and hepatitis, during the data collection were excluded from the study.

## 2.4. Variables and Measurements

The willingness to donate blood was the dependent variable, while socio-demographic factors (age, sex, marital status, and living status), access to the media (radio and TV) and knowledge, and attitude were independent variables.

The willingness of a blood donor is considered voluntary and non remunerated if the adult is giving blood of his/her own free will without receiving payment for it. Replacement donation is when a member of a family is required or when someone donates blood with the intention of replacing the blood their family members have received [26]. Therefore, participants who expressed a willingness to donate blood in the future during the data collection period were assessed using (Yes/No).

Knowledge about the willingness of a participant to donate blood is based on the knowledge questions prepared for participants. Those who scored above the mean (61.8%) for the knowledge questions were labeled as having "good knowledge", while those who scored below the mean were labeled as having "poor knowledge" [20]; regarding their attitude about blood donation, those participants who scored mean (59.0%) and above for the attitude questions were labeled as having a favorable attitude otherwise an "unfavorable attitude" [27]. Social media referred to Facebook and Twitter [28], and mass media referred to television and radio [1].

## 2.5. Sample Size Determinationand Sampling Procedure

Regarding the sample size issue, we calculated the sample size by the single population formula since when we calculated the sample size by the number of possible cases that were included in the sample size calculation for the second objective; it was less than that from the first objective sample size. Therefore, the sample size was calculated by considering the following assumptions: for a single population formula proportion knowledge (16.6%),which studied in Debre Markos town, Northwest Ethiopia [11] using 95% CI and a 4% margin of error. $n = (Z\alpha/2)^2 \times [(p1q1)/(d)]^2$. The sample size was found to be 548 after considering a 10% non-response rate and a 1.5 design effect.

Participants' households were accessed using a multistage sampling technique. From 22 kebeles, 7 kebeles (kebele 1, 3, 6, 8, 13, 16, and 17) were selected by using a simple random sampling technique from the total 22 kebeles. In the case of more than one eligible family member, one eligible adult was taken by a lottery method. Proportion to size allocation was made to determine the required sample size for each randomly selected ketenas. The interval ($K$) value was calculated for each selected ketena by dividing the total households in each selected ketena to the corresponding proportional sample size calculated for each ketena. The initial household was randomly selected by the lottery method. Then, other households were selected at every $K$th interval from a total of 22,809 households. In the event that no eligible adult was identified in a selected household or if the selected household was closed even after revisit, the sampling process continued to the next household until an eligible adult was found (Figure 1).

## 2.6. Data Collection andAnalysis
### 2.6.1. Data Collection, Management, and Quality Assurance

A pretest and structured questionnaire used for this study was developed by reviewing previous related studies after an extensive review of the related literature [29–31], after being validated in the context of the local culture, language, and other factors.

The questionnaire was developed in English, and then translated into Amharic (the local language) and finally back into English to ensure consistency. A pretest was conducted using a 5.0% sample size of the total study sample in Koladiba Town's kebele to establish

the validity and reliability of the questionnaire. The questionnaire was amended based on the findings of the pretest.

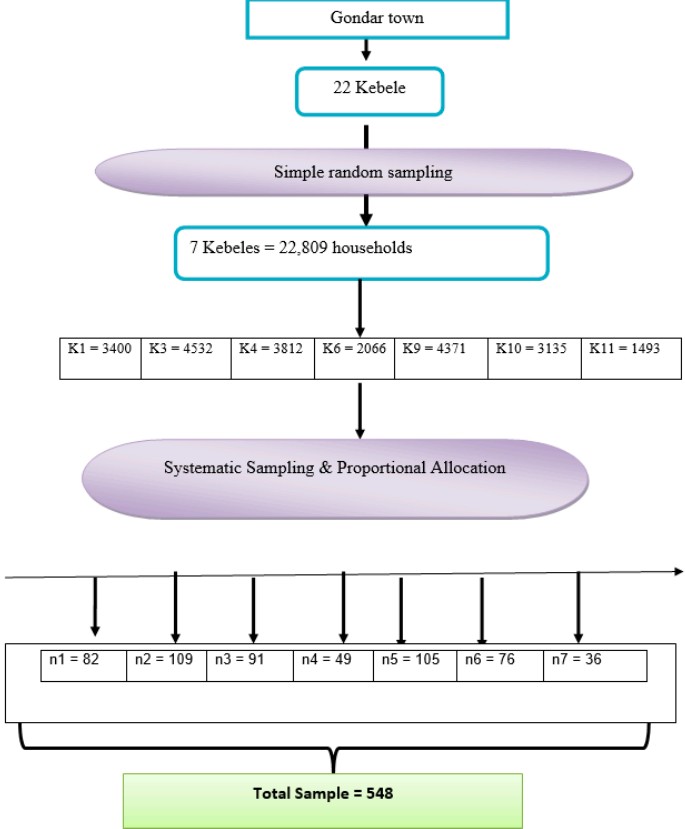

**Figure 1.** Flow chart of the sampling process for estimating the prevalence and associated factors of the willingness to donate blood and its associated factors among adults in Gondar town.

The data collectors were trained on the survey instrument, with an ethical approach to the study participants. The data collectors were four BSc midwives (as data collectors) who administered face-to-face interviews with the study participants.

Besides, completeness and consistency of the questionnaires were reviewed and cross-checked for completeness and consistency by the supervisor, and all the necessary feedback was given to the data collectors immediately. Then, the data were entered using Epi Data version 4.6. Once the data entry was complete, the data were exported to the Statistical Package of the Social Science (SPSS) version 23.0 for data cleaning and analysis. Basic data quality assurance measures were taken, including data cleaning using a browser of the data tables after sorting, graphical exploration of distributions using box plots, histograms, and scatter plots, frequency distributions and cross-tabulations, summary statistics, and statistical outlier detection using sorting. In addition to this, the reliability of the questionnaire was checked by the (Cronbach's alpha 0.83) knowledge and attitudes questionnaires towards a willingness to take part in blood donation.

Descriptive statistics were used for categorical variables, and the mean ± SD (standard deviations) for the continuous variables. Continuous variables were categorized using information from the literature, and categorical variables were recategorized accordingly.

### 2.6.2. Statistical Analysis

Bivariate (crude odds ratio [COR]) and multivariable (adjusted odds ratio [AOR]) values were calculated using logistic regression analysis with 95% confidence interval [CI]. From the bivariate analysis, variables with $p < 0.25$ became candidates for multi-variable analysis. From the multivariable logistic regression analysis, the variables with

a significance level of $p < 0.05$ were taken as statistically significant and independently associated with a willingness for blood donation. The presence of multicollinearity among independent variables was checked using standard error at the cutoff value of 2, and we found a maximum standard error of 1.0, which indicated no multicollinearity. The Hosmer–Lemeshow goodness of fit test was used to check for model fitness by looking at the cut-point $p$-value of >0.05, which had 0.721.

## 3. Results

### 3.1. Descriptive Characteristics

Of the overall sample required (n = 548), 547 of the participants participated in the study with a response rate of 99.1%. The mean age (8.9 ± SD) of the study participants was 32.73 years, and the majority (325 (59.4%)) of study participants were males. Regarding religion, more than half (54.9%) of the study participants were Orthodox Christians. More than three-quarters (76.6%) of the study participants were married. Though the majority (489 (89.7%)) of the study participants had access to social and mass media, only 72 (13.2%) of them had experience in participating in voluntary services (Table 1).

**Table 1.** Sociodemographic characteristics of adult participants living in Gondar town with a willingness to donate blood from Northwest Ethiopia, 2021 (n = 547).

| Variable | Category | Frequency | Percentage |
|---|---|---|---|
| Age in years | 18–25 | 113 | 20.7 |
| | 26–33 | 219 | 40.0 |
| | 34–41 | 131 | 23.9 |
| | ≥41 | 84 | 15.4 |
| Sex | Male | 325 | 59.4 |
| | Female | 222 | 40.6 |
| Living status | Renter | 284 | 51.9 |
| | Rentee | 263 | 48.1 |
| Family income mean ≥2434.01 | <2434.01 | 293 | 53.6 |
| | ≥243.401 | 254 | 46.4 |
| Religion | Orthodox | 300 | 54.8 |
| | Muslim | 185 | 33.8 |
| | Others | 62 | 11.3 |
| Marital status | Married | 419 | 76.6 |
| | Unmarried | 128 | 62.4 |
| Educational status | No formal Education | 99 | 18.1 |
| | Primary school level | 193 | 35.3 |
| | Secondary school level | 94 | 17.2 |
| | Diploma& above | 161 | 29.4 |
| Occupational status | Unemployed | 332 | 60.7 |
| | Employed | 215 | 39.3 |
| Had known medical illness | Yes | 487 | 89.0 |
| | No | 60 | 11.0 |
| Health worker in the family | Yes | 512 | 93.6 |
| | No | 35 | 6.4 |
| Previous participation in voluntary service | Yes | 72 | 13.2 |
| | No | 475 | 86.8 |
| Access to media (social & mass media) | Yes | 487 | 89.0 |
| | No | 60 | 11.0 |

Others = protestant, catholic.

### 3.2. Experience of Participants Blood Transfusion

Of the 547 study participants, the majority (443 (81.0%)) had never experienced blood donation in their lifetime. Only 30.8% of the study participants had ever seen blood being donated in their lifetime (Table 2).

**Table 2.** Experience of blood transfusion among adult participants living in Gondar Town, Northwest Ethiopia, 2021 (n = 547).

| Variable | Category | Frequency | Percentage |
|---|---|---|---|
| Experience of blood donated | Yes | 104 | 19.0 |
| | No | 443 | 81.0 |
| Number of blood donation (n = 104) | Only once | 52 | 48.7 |
| | Two times | 42 | 39.5 |
| | More three times | 10 | 11.8 |
| History of being transfused with blood | Yes | 72 | 18.3 |
| | No | 475 | 81.7 |
| History of blood transfusion in the family | Yes | 178 | 32.5 |
| | No | 369 | 67.5 |
| Member to the Ethiopian Red Cross-society | Yes | 35 | 6.4 |
| | No | 512 | 93.6 |
| Have ever seen blood donated? | Yes | 165 | 30.8 |
| | No | 382 | 69.2 |
| Have ever heard a woman/anyone died because of blood loss? | Yes | 52 | 9.5 |
| | No | 495 | 90.5 |

### 3.3. Knowledge of Participant towards Blood Donation

In this study, the level of participants' knowledge about a willingness to wards blood donation was assessed by 20 items. After calculating the mean values of the study participants, those who had above mean scores were labeled as "good knowledge". Of all the study participants, 338 (61.8%) had "good knowledge"of willingness towards blood donation in the future. Most participants (487 (89.0%)) had never heard of blood donation before this study period. Furthermore, 500 (90.3%) of the study participants did not know about the shortage of blood in the health facilities (Table 3).

**Table 3.** Proportion of knowledge towards a willingness to donate blood among adult participants living in Gondar town, Northwest Ethiopia, 2021 (n = 547).

| Variable | Category | Frequency | % |
|---|---|---|---|
| Had you heard about blood donation before the study? | Yes | 487 | 89.0 |
| | No | 60 | 11.0 |
| What was your source of information? (n = 487) | Television | 59 | 12.1 |
| | Radio | 125 | 25.7 |
| | Social media | 208 | 42.7 |
| | Health worker | 95 | 19.5 |
| What is the age limit for blood donation (lower and upper)? | Yes (18 and 65) | 458 | 83.7 |
| | No | 27 | 4.9 |
| | I don't know | 62 | 11.3 |

**Table 3.** *Cont.*

| Variable | Category | Frequency | % |
|---|---|---|---|
| What is the minimum weight to donate blood? | Yes (≥45 Kg) | 405 | 74.0 |
| | No | 57 | 10.4 |
| | I don't know | 85 | 15.5 |
| Do you know blood donation is not harmful? | Yes | 454 | 83.0 |
| | No | 93 | 17.0 |
| Do you know blood can be donated periodically? | Yes | 87 | 19.5 |
| | No | 460 | 80.5 |
| How frequently can a person donate blood? (In month) | Once per three | 46 | 8.4 |
| | Twice per three | 41 | 7.5 |
| Do you know if blood transfusion can transmit infections? | Yes | 195 | 35.6 |
| | No | 352 | 64.4 |
| What are blood transfusions related infections (n = 195)? | HIV/AIDS | 11 | 5.9 |
| | Hepatitis | 40 | 21.3 |
| | Malaria | 73 | 38.8 |
| | Others | 64 | 34.0 |
| Do you know any pregnant women who loss blood due to bleeding and needed a blood transfusion? | Yes | 65 | 11.9 |
| | No | 482 | 88.1 |
| Do you know any patients who had been in a severe accident with injuries that needed a blood transfusion? | Yes | 37 | 6.8 |
| | No | 510 | 93.2 |
| Do you know any anemic patients due to malaria who needed a blood transfusion? | Yes | 521 | 95.2 |
| | No | 26 | 4.8 |
| Do you know any places that facilitate blood donation? | Yes | 65 | 11.9 |
| | No | 482 | 88.1 |
| Do you know there is shortage of blood in the health facilities? | Yes | 47 | 9.7 |
| | No | 500 | 90.3 |
| Do you know if blood donation should be voluntarily? | Yes | 452 | 82.6 |
| | No | 95 | 17.4 |
| Do you know that blood donation is important to donors? | Yes | 133 | 24.3 |
| | No | 414 | 75.7 |
| Knowledge | Good knowledge (≥20.11170) | 338 | 61.8 |
| | Poor Knowledge | 209 | 38.2 |

Notes: HIV/AIDS, Human immunodeficiency virus/acquired immunodeficiency syndrome.

### 3.4. Attitude of Participants towards Willingness to Donate Blood

Among the 547 study participants, nearly two-thirds (59.0%) had a favorable attitude toward blood donation, while41% of the study participants had an unfavorable attitude. Furthermore, 333 (60.9%) of the study participants agreed that all eligible people should donate blood (Table 4).

**Table 4.** Participants attitude towards blood donation among adult participants living in Gondar town 2021 (n = 547).

| Variables | Category | Frequency | Percentage |
|---|---|---|---|
| Do you agree that all people should donate blood if he/she is eligible? | Strongly agree | 333 | 60.9 |
| | Agree | 18 | 3.3 |
| | Neutral | 2 | 0.4 |
| | Disagree | 16 | 2.9 |
| | Strongly disagree | 178 | 32.5 |
| Do you agree that blood should be donated to family members only? | Strongly agree | 62 | 11.5 |
| | Agree | 47 | 8.6 |
| | Neutral | 41 | 7.5 |
| | Disagree | 59 | 10.8 |
| | Strongly disagree | 338 | 61.8 |
| Do you agree that donation is harm full to donors? | Strongly agree | 320 | 58.5 |
| | Agree | 31 | 5.7 |
| | Neutral | 40 | 7.3 |
| | Disagree | 63 | 11.5 |
| | Strongly disagree | 93 | 17.0 |
| Do you agree that blood donation should be voluntary? | Strongly agree | 66 | 12.1 |
| | Agree | 134 | 24.4 |
| | Neutral | 11 | 2.0 |
| | Disagree | 76 | 13.9 |
| | Strongly disagree | 260 | 47.5 |
| Do you agree that financial compensation for blood donation is appropriate? | Strongly agree | 361 | 66.0 |
| | Agree | 36 | 6.6 |
| | Neutral | 21 | 3.8 |
| | Disagree | 24 | 4.4 |
| | Strongly disagree | 105 | 19.2 |
| Do you agree that blood should be donated continuously? | Strongly agree | 66 | 12.1 |
| | Agree | 134 | 24.5 |
| | Neutral | 11 | 2.0 |
| | Disagree | 76 | 13.9 |
| | Strongly disagree | 260 | 47.5 |
| Do you agree that there is a shortage of blood in health facilities? | Strongly agree | 84 | 15.5 |
| | Agree | 37 | 6.8 |
| | Neutral | 18 | 3.3 |
| | Disagree | 28 | 5.1 |
| | Strongly disagree | 380 | 69.5 |

### 3.5. Prevalence of Willingness to Donate Blood

Of the 547 respondents, below half (45.3%) were willing to donate blood voluntarily in the future, while the remaining 54.8% were unwilling for a variety of reasons. From this, the reasons mentioned for not donating blood among the nondonors were fear of having health risks after donation (18 (3.3%)), fear of being sick (83 (15.2%)), fear of weight loss

(93 (17.0%)), lack of information on where, when, and how to donate blood (52 (9.5%)) and other reasons (chronic diseases like diabetic mellitus and hypertension) (53 (9.7%)).

### 3.6. Factors Associated with Willingness to Donate Blood

A bivariate logistic regression analysis revealed that marital status, occupation, living status, having a history of previous blood donations, having seen the need for donation when a deceased family member died through blood loss, and attitude and knowledge were predictor variables. However, in multivariable binary logistic regression analysis, the renter's living status, history of blood donation, having seen the need for donation when a deceased family member died through blood loss, and participants having good knowledge remained significant factors for willingness to donate blood.

As a result, those study participants who lived in their own households were 3.19 times more likely to be willing to donate blood than those who were renting their accommodation [AOR: 3.19; 95% CI: 2.09, 4.62]. The respondents with a history of blood donation practice were 1.90 times more likely to be willing to donate blood than their counterparts [AOR: 1.90; 95% CI: 1.16, 3.19]. Likewise, the study participants who had seen blood donated were 2.56 times more likely to donate blood voluntarily than those participants who had never seen blood donated [AOR: 2.56; 95% CI: 1.65, 6.95]. Study participants who had a history of dead relatives with blood loss were 2.28 times more likely to have willingness towards blood donation than those respondents who had no history of dead relatives through blood loss [AOR: 2.28; 95% CI:1.19, 4.36]. Furthermore, those respondents who had good knowledge about the importance of blood donation were 2.23 times more likely to practice blood donation than those who had poor knowledge [AOR: 2.23; 95% CI: 1.49, 3.34] (It refers to (Table 5).

**Table 5.** Bivariate and multivariable logistic regression of factors associated with a willingness to donate blood among adult participants in Gondar town, Northwest Ethiopia, 2021 (n = 547).

| Variables | Willing to Donate Blood | | COR 95% CI | AOR 95% CI | *p*-Value |
|---|---|---|---|---|---|
| | **Yes** | **No** | | | |
| Marital status | | | | | |
| Unmarried | 69 | 59 | 1.57 (1.05, 2.33) | 1.23 (0.78, 1.92) | 0.342 |
| Married | 179 | 240 | Ref. | Ref. | |
| Occupation | | | | | |
| Employed | 112 | 113 | 1.36 (1.11, 2.21) | 1.39 (0.95, 2.05) | 0.434 |
| Un employed | 136 | 186 | Ref. | Ref. | |
| Living status | | | | | |
| Renter | 133 | 151 | 1.13 (2.87, 5.91) | 3.19 (2.09, 4.62) *** | 0.000 |
| Rentee | 115 | 148 | Ref. | Ref. | |
| History of donate blood | | | | | |
| Yes | 45 | 59 | 0.90 (1.68, 4.20) | 1.90 (1.16, 3.19) * | 0.003 |
| No | 203 | 240 | Ref. | Ref. | |
| Ever seen blood donated | | | | | |
| Yes | 118 | 47 | 2.26 (1.88, 4.13) | 2.56 (1.65, 6.95) *** | 0.000 |
| No | 201 | 181 | Ref. | Ref. | |
| Death family with blood lose | | | | | |
| Yes | 61 | 16 | 3.91 (2.08, 6.63) | 2.28 (1.19, 4.36) * | 0.001 |
| No | 232 | 238 | Ref. | Ref. | |
| Knowledge | | | | | |
| Good | 109 | 100 | 1.56 (1.10, 2.21) | 2.23 (1.49, 3.34) *** | 0.000 |
| Poor | 139 | 199 | Ref. | Ref. | |
| Attitude | | | | | |
| Favorable | 176 | 157 | 2.23 (1.55, 3.16) | 1.39 (0.89, 2.59) | 0.231 |
| Unfavorable | 72 | 142 | Ref. | Ref. | |

* $p < 0.05$; *** $p < 0.001$.

## 4. Discussion

Blood transfusion has become the most effective treatment for saving patients who had organ failure [32]. Healthcare providers are always eager to donate blood; however, the supply and demand for blood are vastly different [33]. Thus, the study assessed the prevalence of a willingness to donate blood and its determinants at the community level among adults living in Gondar town, Northwest Ethiopia.

The overall prevalence of a willingness to donate blood among adults was found to be 45.3%, with a 95% CI (41.4, 49.9%). This result is lower than the studies conducted among health professionals in Gondar, Ethiopia, where the value was 74.6–78.1% [14,16]. In Jimma, Ethiopia [17], it was 58.1%, and in Nigeria, it was 59.3–73% [15,34]. The possible reasons might be due to the difference in awareness and sociodemographic characteristics between the study participants. The current study was a community-based study, where the study participants' willingness might have been less due to a lack of awareness of the importance of blood donation practices when compared to the previous studies conducted among health professionals and medical student participants. In this study, about 99 (18.1%) of the study participants were not attending formal education, whereas 100% of the study participants in Jimma, Ethiopia, were medical students. It is expected that those individuals with a higher level of education will have better knowledge and a positive attitude towards blood donation, thereby being willing to donate blood. Thus, as the level of education increases, participants' knowledge of blood donation also increases.

However, the prevalence of a willingness to donate blood in this study is similar to the community-based studies conducted in Adama town, Oromia region, Ethiopia (39.8.0%) [35] and in the Philippines, where 48.0% [36] of the study participants were willing to donate blood. The possible explanations might be due to the similarity of the study participants' knowledge and sociodemographic characteristics. Around 61.8% of the study participants in this study had good levels of knowledge of blood donation, and 48.5% of the study participants in the Oromia regional state, central Ethiopia, had good knowledge of donated blood saving lives. Ensuring the health and well-being of the population is one of the global agendas, for which national and international organizations give emphasis. For this reason, the public should be encouraged to donate blood regularly so as to obtain are liable amount of blood for the blood bank.

In the current study, the multivariate logistic regression revealed that those who rented households out to others were 3.19 times more likely to be willing to donate blood in the future than those rent a house from someone else. This might be due to the fact that the rented who permanently lived in the town, had more of a sense of the shortage of blood in the study area. The current study, like another study conducted in Saudi Arabia [30], found that participants with a history of blood donation were 1.90 times more likely to be willing to donate blood in the future. This might be due to the fact that the previous blood donated may increase their level of satisfaction (for potentially saving lives) and results in a change of behavior towards blood donation. This finding is supported by other studies from a different part of Ethiopia [11,22,37], where those participants that had experienced blood donation could motivate other individuals to donate blood to save lives. Those participants who have experience in blood donation are very important for sharing information with other eligible adults.

In this study, those study participants who had seen others donate blood in their lifetime were 2.56 more likely to be willing to donate blood compared to those who hadn't seen blood being donated. This might be due to the study participants' motivation to donate blood as a result of seeing others donating blood, which might reduce unreasonable fears of being an anemic patient and misunderstandings about the side effects of blood donation. This is in line with a study carried out in Dessie Town [22], where the participants who had seen others donate blood were significantly associated with a willingness to donate blood. Providing blood donation campaigns for potential donors about the health benefits of donating blood, the volume of blood donated, and the number of adults benefiting from

a single unit of blood donated is very important when improving the number of willing participants towards blood donation.

In the present study, the study participants who had a family history of blood transfusion had higher odds of being willing to donate blood compared with their counterparts. The possible important explanation is that those individuals with a family that has a history of blood donation may have received blood from a volunteer blood donor and may have wanted to donate blood as soon as they understood that giving blood saves many lives.

Likewise, study participants who had lost relatives to bloodlosswere 2.28 times more likely to show willingness toward blood donation than their counterparts. This is because having a deceased relative (in this case, a deceased relative through blood loss) may have had a powerful role in increasing their willingness compared to participants who had no deceased relatives through blood loss. This could be explained by the fact that people will have an increased willingness to donate blood after experiencing that a family member with health problems required a blood transfusion.

Moreover, participants who had good knowledge of blood donation were 2.23 times more willing to give blood in the future compared with those having poor knowledge. This can be explained by the fact that those participants who were knowledgeable about blood donation knew that donating blood does not harm the donor and, hence, could have donated blood to save lives. The other perspectives might be due to the participant's knowledge of the health benefits of donating blood in terms of reducing the risk of cancer and heart attack. This explanation is supported by the study conducted among high school students in Addis Ababa [38]. Even though donating blood voluntarily in order to get enough blood from the blood bank is not required in our setting, most of the time, people donate with the intention of replacing blood given to their loved ones. However, the public should be encouraged to donate blood voluntarily so as to get safe and reliable blood. Our intervention to increase blood donation practice should focus on improving the knowledge level of adults and also increasing the accessibility to blood bank services; approaching potential donors is essential.

Common reasons reported for people not being willing to donate blood include fear of health problems after screening (18 (3.3%)), fear of pain (83 (15.2%)), fear of weight loss (93 (17.0%)), lack of information (52 (9.5%)), and other reasons (diabetic mellitus and hypertension) (53 (9.7%)), which is consistent with the study conducted in different parts of Ethiopia and India [4,26,39]. Participants who have knowledge about the health benefits of donating blood, the volume of blood donated, and the number of patients benefiting from a single unit of blood donated are very important with regards to increasing the number of adults willing to donate blood.

*Limitations of the Study*

We acknowledge some important possible limitations that should be considered when interpreting our results. First, the study was cross-sectional, which is a design that does not establish cause–effect relationships. Second, social desirability and recall bias might be introduced. A final limitation of the study is the lack of qualitative supporting data; this may have provided some answers about why some people are not willing to donate blood.

## 5. Conclusions

In this study, the willingness to donate blood among adult participants at the community level was found to be low. The main reasons for the unwillingness of the participants to donate blood are an unreasonable fear of being anemic patients and misunderstandings about the amount of blood donated. To avert this problem, healthcare providers, national blood banks, and transfusion agencies should design effective strategies to promote and motivate these types of communities. In addition to this, participants should receive information about the health benefits of donating blood, the volume of blood donated, and the number of patients benefiting from a single unit of blood donated.

Regarding other factors, participants who live in rented accommodation, have a history of previous blood donation, have seen blood being donated, have deceased relatives through blood loss causes, and have good knowledge are the most willing to donate blood in the future, with these factors being the most important predictors. Moreover, the newer research questions generated through this study need to be included in further research on this topic. Besides, large-scale, in-depth behavioral studies need to be conducted to explore the distal and proximal societal factors that affect communities' willingness towards blood donation.

**Author Contributions:** All stated authors (A.M.Z. and Z.N.A.) were involved in the study from its inception to design, acquisition of data, analysis and interpretation, and drafting of the manuscript. All authors have read and agreed to the published version of the manuscript.

**Funding:** This research received no external funding.

**Institutional Review Board Statement:** The study was conducted in accordance with the Declaration of Helsinki, and approved by the Ethics Committee of Real Dream Health Sciences College with ethical letter protocol number: RDC/20/09/2021.

**Informed Consent Statement:** In addition, after explaining the importance of the study, permission letters were taken from each of the kebeles/ketenas administrators, and verbal informed consent was obtained from all participants. Names or specific addresses of the study participants were coded, and confidentiality was assured. Their rights to refuse to participate, to refuse to answer any or all questions, and to leave the interview at any time were respected. Patients and/or the public were not involved in the design, conduct, reporting, or dissemination plans of this research.

**Data Availability Statement:** The data sets used and/or analyzed during this study are available from the corresponding author on reasonable request.

**Acknowledgments:** Firstly: the authors would like to thank the real Dream University of College for the approval of the ethical clearance. Secondly, we would like to acknowledge the study participants, data collectors, and supervisors who participated in the study. Lastly, we want to appreciate our colleagues and friends for giving us support and constructive comments.

**Conflicts of Interest:** The authors have as declare no conflict of interest.

## Abbreviations

| | |
|---|---|
| AOR | adjusted odds ratio |
| COR | crude odds ratio |
| HCW | health care workers |
| SPSS | Statistical Packages for Social Sciences |
| SSA | Sub-Saharan Africa |
| WHO | World Health Organization |

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
