# Peer review of "Willingness and Its Associated Factors for Blood Donation in Gondar Town, Northwest Ethiopia: A Community-Based Cross-Sectional Study"

_2673-947X, doi:10.3390/hygiene2040019_

Round 1

Reviewer 1 Report (New Reviewer)

Overall, an interesting paper with finding relevant to a current blood shortage crisis. The main issue with this work is the language. For example; "Willingness to donate blood" not "willingness to blood donation". I suggest having someone review this prior to resubmission. Some further points to address are below. 

Lines 22-25: Try to be more broad about your definitions in your opening sentences. Rather than providing specific examples of blood loss, I would just say "trauma, long-term therapies and medical or haematological conditions". 

Line 32: When were these blood donations? No reference. 

Line 47: Health professionals? This should be defined. 

Lines 62-64: I am not sure that the latitude or longitude or elevation of the town is important in this study. 

Lines 71-73: Please be specific about your inclusion and exclusion criteria. What types of conditions excluded people from being involved and why. 

Line 161: Should be three-quarters (not third-fourths)

It would be beneficial to divide Table 1 and Table 2 up to show the differences in age or gender between the variables. Therefore, have a column for males and a column for females OR a column for each age group. Then look at whether there are any statistically significant differences. This would help to target future promotion. 

Table 5 has some really valuable information but needs to be formatted correctly so that the meaning is not lost. See the table formatting guidelines.

All tables could be better formatted.  

A further limitation of the study is the lack of qualitative supporting data. This may have provided some answers around why some people were more likely to donate blood. 

The discussion is fair although some higher level thinking would be beneficial. Try to tie each component of your results together rather than just saying what you found and a possible reason for it. 

Author Response

2022-9-23

Dear; Editor and Reviewers

Please accept our revised manuscript and note our point-by point response to reviewers below for the manuscript titled “Willingness and its factors with blood donation in Gondar Town, Northwest Ethiopia. A community-based cross-sectional study

Our revised manuscript continues to meet the journal’s formal requirements, including the overall word count.

We wish to express our sincere appreciation to the Editorials and the reviewers for their invaluable inputs; we believe we have now strengthened our paper. Please see below a point by point response to the comments raised.

N.B Authors’ responses are indicated in bold italics

Reviewer 1 and 2: Overall, an interesting paper with finding relevant to a current blood shortage crisis.

The main issue with this work is the language. For example; "Willingness to donate blood" not "willingness to blood donation". I suggest having someone review this prior to resubmission. Some further points to address are below. 

Lines 22-25: Try to be more broad about your definitions in your opening sentences. Rather than providing specific examples of blood loss, I would just say "trauma, long-term therapies and medical or haematological conditions". 

Authors’ response: Thank you very much for this comment and the concern is noted and well taken and added trauma, long-term therapies and medical or haematological conditions in the study

Line 32: When were these blood donations?  No reference. 

Authors’ response: Thank you very much for this comment and the concern is noted and we added the reference

Line 47: Health professionals? This should be defined. 

Authors’ response: Thank you for this positive comment. We appreciate the reviewer’s need of clarity and in that study Health professionals means medical doctor, nurse and midwife.

Lines 62-64: I am not sure that the latitude or longitude or elevation of the town is important in this study.

Authors’ response: We appreciate this observation. We have now made the necessary changes which the latitude or longitude or elevation of the town is removed.

 Lines 71-73: Please be specific about your inclusion and exclusion criteria. What types of conditions excluded people from being involved and why. Line .161: Should be three-quarters (not third-fourths)

Authors’ response: We greatly appreciate the reviewer’s efforts to carefully review the paper and the valuable suggestions offered. We appreciate the reviewer’s suggestion and we have now corrected by adding inclusion and exclusion criteria in the revised manuscript.

It would be beneficial to divide Table 1 and Table 2 up to show the differences in age or gender between the variables. Therefore, have a column for males and a column for females OR a column for each age group. Then look at whether there are any statistically significant differences. This would help to target future promotion. 

Table 5 has some really valuable information but needs to be formatted correctly so that the meaning is not lost. See the table formatting guidelines.

All tables could be better formatted.  

Authors’ response: This concern is noted and well taken. We have now revised the table and added the references. We tried to correct the based on guidelines. We appreciate this observation. We have now made the necessary changes and chose to include the information requested by the reviewer in the whole Tables.   

A further limitation of the study is the lack of qualitative supporting data. This may have provided some answers around why some people were more likely to donate blood. 

Authors’ response: This request is well taken. We also appreciate the fact that reviewer has acknowledged the inclusion of the lack of qualitative supporting data as our study limitations.

The discussion is fair although some higher level thinking would be beneficial. Try to tie each component of your results together rather than just saying what you found and a possible reason for it. 

Authors’ response: We greatly appreciate the reviewer’s efforts to carefully review the paper and the valuable suggestions offered. We appreciate the reviewer’s suggestion and we have now corrected line by line in the whole the result and discussion parts in revised manuscript.

Finally, We wish to express our sincere appreciation to the invaluable inputs given us, we believe we have now strengthened our paper and here we are in a position to reassure you that we have considered language revision and the manuscript has been reviewed meticulously by a native English speaker, whose name is Solomon Girma, who has 11 years of experience as a lecturer of English language and literature in university of Gondar, Ethiopia. Mr Girma is also an assistant instructor of Germen as a foreign language in University of Gondar, Ethiopia. And also necessary corrections made.

Reviewer 2 Report (New Reviewer)

I would like to thank the editors of this journal for the opportunity to give my point of view on this interesting research project. On the other hand, I would like to congratulate the authors of this research. I believe that it is very necessary to investigate the attitude and knowledge that citizens have about donating blood, especially in regions where there is a shortage of blood. These types of studies help with their results to create new methodologies to address systems that facilitate the proper functioning of health institutions and better treatment of patients.

I believe that the study is well thought out and the methodology is appropriate, but I would like to make some comments that I think could be addressed by the authors and improve some aspects of the study.

The study deals with patient information, it would be appropriate to dedicate a paragraph in the material and methods where the work done with the ethics committee is indicated.

Although they are raised in the discussion, it would be interesting to group the limitations of the study in a paragraph in this section, as well as future applications and uses of the results of the study.

In this study, the information that the citizen has received about blood donation is very important. I think it would be interesting to indicate in the introduction the information campaigns that have been carried out in this regard in this area, giving priority to those carried out in social media. The results show that respondents receive information mainly through social networks, I think this is so important that it should have been possible to indicate which social network the information was received through and whether this information was shared... in this sense, the study could have been improved.

In section 2.4 you interpret results. The results should be in the results section and the interpretation in the discussion.

The results section is full of interpretations of the data, I think that the results section should only show results, the interpretations should be in the discussion section.

Several studies, especially those related to organ donation, make a good distinction between attitude towards organ donation and knowledge, we can have an adequate knowledge and a negative attitude. Here are some reference studies.

DOI: 10.1097/TP.0b013e318231ea17
https://doi.org/10.3390/ijerph19148524
 DOI: 10.1034/j.1600-6143.2001.010114.x
10.1016/j.transproceed.2019.09.020
doi:http://dx.doi. org/10.1016/j.pec.2017.12.019

Again, I congratulate the researchers for the effort and dedication they have put into this great study and thank the editors of the journal for the opportunity to provide my opinion on the research.

Author Response

2022-9-23

Dear; Editor and Reviewers

Please accept our revised manuscript and note our point-by point response to reviewers below for the manuscript titled “Willingness and its factors with blood donation in Gondar Town, Northwest Ethiopia. A community-based cross-sectional study

Our revised manuscript continues to meet the journal’s formal requirements, including the overall word count.

We wish to express our sincere appreciation to the Editorials and the reviewers for their invaluable inputs; we believe we have now strengthened our paper. Please see below a point by point response to the comments raised.

N.B Authors’ responses are indicated in bold italics

Reviewer 1 and 2: Overall, an interesting paper with finding relevant to a current blood shortage crisis.

The main issue with this work is the language. For example; "Willingness to donate blood" not "willingness to blood donation". I suggest having someone review this prior to resubmission. Some further points to address are below. 

Lines 22-25: Try to be more broad about your definitions in your opening sentences. Rather than providing specific examples of blood loss, I would just say "trauma, long-term therapies and medical or haematological conditions". 

Authors’ response: Thank you very much for this comment and the concern is noted and well taken and added trauma, long-term therapies and medical or haematological conditions in the study

Line 32: When were these blood donations?  No reference. 

Authors’ response: Thank you very much for this comment and the concern is noted and we added the reference

Line 47: Health professionals? This should be defined. 

Authors’ response: Thank you for this positive comment. We appreciate the reviewer’s need of clarity and in that study Health professionals means medical doctor, nurse and midwife.

Lines 62-64: I am not sure that the latitude or longitude or elevation of the town is important in this study.

Authors’ response: We appreciate this observation. We have now made the necessary changes which the latitude or longitude or elevation of the town is removed.

 Lines 71-73: Please be specific about your inclusion and exclusion criteria. What types of conditions excluded people from being involved and why. Line .161: Should be three-quarters (not third-fourths)

Authors’ response: We greatly appreciate the reviewer’s efforts to carefully review the paper and the valuable suggestions offered. We appreciate the reviewer’s suggestion and we have now corrected by adding inclusion and exclusion criteria in the revised manuscript.

It would be beneficial to divide Table 1 and Table 2 up to show the differences in age or gender between the variables. Therefore, have a column for males and a column for females OR a column for each age group. Then look at whether there are any statistically significant differences. This would help to target future promotion. 

Table 5 has some really valuable information but needs to be formatted correctly so that the meaning is not lost. See the table formatting guidelines.

All tables could be better formatted.  

Authors’ response: This concern is noted and well taken. We have now revised the table and added the references. We tried to correct the based on guidelines. We appreciate this observation. We have now made the necessary changes and chose to include the information requested by the reviewer in the whole Tables.   

A further limitation of the study is the lack of qualitative supporting data. This may have provided some answers around why some people were more likely to donate blood. 

Authors’ response: This request is well taken. We also appreciate the fact that reviewer has acknowledged the inclusion of the lack of qualitative supporting data as our study limitations.

The discussion is fair although some higher level thinking would be beneficial. Try to tie each component of your results together rather than just saying what you found and a possible reason for it. 

Authors’ response: We greatly appreciate the reviewer’s efforts to carefully review the paper and the valuable suggestions offered. We appreciate the reviewer’s suggestion and we have now corrected line by line in the whole the result and discussion parts in revised manuscript.

Finally, We wish to express our sincere appreciation to the invaluable inputs given us, we believe we have now strengthened our paper and here we are in a position to reassure you that we have considered language revision and the manuscript has been reviewed meticulously by a native English speaker, whose name is Solomon Girma, who has 11 years of experience as a lecturer of English language and literature in university of Gondar, Ethiopia. Mr Girma is also an assistant instructor of Germen as a foreign language in University of Gondar, Ethiopia. And also necessary corrections made.

Round 2

Reviewer 1 Report (New Reviewer)

Thank you for addressing the previous comments. The revisions have improved the paper overall and the flow of information is much better. One minor concern in the abstract:

"However, there is paucity evidence on assessing donate blood willingness" does not make sense and should be reworded. 

Author Response

Dear; Editor and Reviewers

Please accept our revised manuscript and note our point-by point response to reviewers below for the manuscript titled “Willingness and its factors with blood donation in Gondar Town, Northwest Ethiopia. A community-based cross-sectional study

Our revised manuscript continues to meet the journal’s formal requirements, including the overall word count.

We wish to express our sincere appreciation to the Editorials and the reviewers for their invaluable inputs; we believe we have now strengthened our paper. Please see below a point by point response to the comments raised.

N.B Authors’ responses are indicated in bold italics

Reviewer 1: The need of clarity of adults can serve as an essential pool for potential blood donors in the abstract part. However, there is paucity evidence on assessing donate blood willingness.

Authors’ response: Thank you very much for this novel comment and the concern is noted and corrected “Though World Health Organization recommends 100% willingness for blood donation, the percentage of blood collected from willing blood donors and the average annual blood collection rate are extremely low in Ethiopia. Adults can serve as an essential pool to meet the demand of safe blood”. Authors’ response: We appreciate this observation. This concern is noted and well taken. We have now revised the whole manuscript based on guidelines. We have now made the necessary changes and chose to include the information requested by the reviewer in the whole Tables. Please check from track change  

Finally, we wish to express our sincere appreciation to the invaluable inputs given us, we believe we have now strengthened our paper and here we are in a position to reassure you that again we have considered language revision and the manuscript has been reviewed meticulously by a native English speaker, whose name is Solomon Girma, who has 11 years of experience as a lecturer of English language and literature in university of Gondar, Ethiopia. Mr Girma is also an assistant instructor of Germen as a foreign language in University of Gondar, Ethiopia. And also necessary corrections made.

Reviewer 2: The study deals with patient information, it would be appropriate to dedicate a paragraph in the material and methods where the work done with the ethics committee is indicated.

Authors’ response: Thank you very much for this comment. We appreciate the reviewer’s suggestion and the concern is noted.We greatly appreciate the reviewer’s efforts to carefully review the paper and the valuable suggestions offered.

Although they are raised in the discussion, it would be interesting to group the limitations of the study in a paragraph in this section, as well as future applications and uses of the results of the study.

Authors’ response: Thank you very much for this comment and the concern is noted and we revised the discussion part. Please check from track change

In this study, the information that the citizen has received about blood donation is very important. I think it would be interesting to indicate in the introduction the information campaigns that have been carried out in this regard in this area, giving priority to those carried out in social media. The results show that respondents receive information mainly through social networks, I think this is so important that it should have been possible to indicate which social network the information was received through and whether this information was shared... in this sense, the study could have been improved.

Authors’ response: We are grateful for providing us a second chance to consider our manuscript, and we also very much appreciate your suggestions, which have been very helpful in improving the manuscript. We also thank the reviewers for their careful reading of our text. We tried to correct the manuscript us our understanding.

In section 2.4 you interpret results. The results should be in the results section and the interpretation in the discussion.
The results section is full of interpretations of the data, I think that the results section should only show results, the interpretations should be in the discussion section.
Authors’ response: We appreciate this observation. We have now made the necessary changes and chose to include the information requested by the reviewer in the results section should only show results and interpretations should be in the discussion section.

Several studies, especially those related to organ donation, make a good distinction between attitude towards organ donation and knowledge, we can have an adequate knowledge and a negative attitude. Here are some reference studies.

Authors’ response: Thank you for this positive comment. We appreciate the reviewer’s suggestion and we have now corrected in the revised manuscript. Line 35 to 35(According to the study done in Jordan poor knowledge and negative attitude of blood donors were barriers for blood donation[18].

Finally, we wish to express our sincere appreciation to the invaluable inputs given us, we believe we have now strengthened our paper and here we are in a position to reassure you that again we have considered language revision and the manuscript has been reviewed meticulously by a native English speaker, whose name is Solomon Girma, who has 11 years of experience as a lecturer of English language and literature in the university of Gondar, Ethiopia. Mr. Girma is also an assistant instructor of German as a foreign language at the University of Gondar, Ethiopia. And also necessary corrections made.

Reviewer 2 Report (New Reviewer)

I am sorry but the authors have not made any of my corrections or suggestions, there are no replies in the system. I have read the considerations made by my colleague reviewer 1 and I consider them very timely.

Author Response

Dear; Editor and Reviewers

Please accept our revised manuscript and note our point-by point response to reviewers below for the manuscript titled “Willingness and its factors with blood donation in Gondar Town, Northwest Ethiopia. A community-based cross-sectional study

Our revised manuscript continues to meet the journal’s formal requirements, including the overall word count.

We wish to express our sincere appreciation to the Editorials and the reviewers for their invaluable inputs; we believe we have now strengthened our paper. Please see below a point by point response to the comments raised.

N.B Authors’ responses are indicated in bold italics

Reviewer 1: The need of clarity of adults can serve as an essential pool for potential blood donors in the abstract part. However, there is paucity evidence on assessing donate blood willingness.

Authors’ response: Thank you very much for this novel comment and the concern is noted and corrected “Though World Health Organization recommends 100% willingness for blood donation, the percentage of blood collected from willing blood donors and the average annual blood collection rate are extremely low in Ethiopia. Adults can serve as an essential pool to meet the demand of safe blood”. Authors’ response: We appreciate this observation. This concern is noted and well taken. We have now revised the whole manuscript based on guidelines. We have now made the necessary changes and chose to include the information requested by the reviewer in the whole Tables. Please check from track change  

Finally, we wish to express our sincere appreciation to the invaluable inputs given us, we believe we have now strengthened our paper and here we are in a position to reassure you that again we have considered language revision and the manuscript has been reviewed meticulously by a native English speaker, whose name is Solomon Girma, who has 11 years of experience as a lecturer of English language and literature in university of Gondar, Ethiopia. Mr Girma is also an assistant instructor of Germen as a foreign language in University of Gondar, Ethiopia. And also necessary corrections made.

Reviewer 2: The study deals with patient information, it would be appropriate to dedicate a paragraph in the material and methods where the work done with the ethics committee is indicated.

Authors’ response: Thank you very much for this comment. We appreciate the reviewer’s suggestion and the concern is noted.We greatly appreciate the reviewer’s efforts to carefully review the paper and the valuable suggestions offered.

Although they are raised in the discussion, it would be interesting to group the limitations of the study in a paragraph in this section, as well as future applications and uses of the results of the study.

Authors’ response: Thank you very much for this comment and the concern is noted and we revised the discussion part. Please check from track change

In this study, the information that the citizen has received about blood donation is very important. I think it would be interesting to indicate in the introduction the information campaigns that have been carried out in this regard in this area, giving priority to those carried out in social media. The results show that respondents receive information mainly through social networks, I think this is so important that it should have been possible to indicate which social network the information was received through and whether this information was shared... in this sense, the study could have been improved.

Authors’ response: We are grateful for providing us a second chance to consider our manuscript, and we also very much appreciate your suggestions, which have been very helpful in improving the manuscript. We also thank the reviewers for their careful reading of our text. We tried to correct the manuscript us our understanding.

In section 2.4 you interpret results. The results should be in the results section and the interpretation in the discussion.
The results section is full of interpretations of the data, I think that the results section should only show results, the interpretations should be in the discussion section.
Authors’ response: We appreciate this observation. We have now made the necessary changes and chose to include the information requested by the reviewer in the results section should only show results and interpretations should be in the discussion section.

Several studies, especially those related to organ donation, make a good distinction between attitude towards organ donation and knowledge, we can have an adequate knowledge and a negative attitude. Here are some reference studies.

Authors’ response: Thank you for this positive comment. We appreciate the reviewer’s suggestion and we have now corrected in the revised manuscript. Line 35 to 35(According to the study done in Jordan poor knowledge and negative attitude of blood donors were barriers for blood donation[18].

Finally, we wish to express our sincere appreciation to the invaluable inputs given us, we believe we have now strengthened our paper and here we are in a position to reassure you that again we have considered language revision and the manuscript has been reviewed meticulously by a native English speaker, whose name is Solomon Girma, who has 11 years of experience as a lecturer of English language and literature in the university of Gondar, Ethiopia. Mr. Girma is also an assistant instructor of German as a foreign language at the University of Gondar, Ethiopia. And also necessary corrections made.

Round 3

Reviewer 2 Report (New Reviewer)

I believe that the authors have made significant changes to the manuscript and heeded my suggestions.

This manuscript is a resubmission of an earlier submission. The following is a list of the peer review reports and author responses from that submission.

Round 1

Reviewer 1 Report

Dear Author,

Title of your study is indeed cardinal to transfusion services. Study is developed systematically and questionnaire and results are also explained in detail, however there are multiple mistakes in sentence making, grammar and word choice. I have attached your manuscript file with highlighted mistakes.

Regards.

Author Response

Dear; Editor and Reviewers

Please accept our revised manuscript and note our point-by point response to reviewers below for the manuscript titled “Willingness and its factors with blood donation in Gondar Town, Northwest Ethiopia. A community-based cross-sectional study

Our revised manuscript continues to meet the journal’s formal requirements, including the overall word count.

We wish to express our sincere appreciation to the Editorials and the reviewers for their invaluable inputs; we believe we have now strengthened our paper. Please see below a point by point response to the comments raised.

N.B Authors’ responses are indicated in bold italics

  1. Reviewer 1: In the Introduction part: The quality of the English used throughout your manuscript does not currently meet our requirements, as there are several incorrect sentence constructions and grammatical errors throughout obscuring the message the authors want to convey. We recommend that you ask a native English speaking colleague to help you copy-edit the paper. If this is not possible, you may need to use a professional language editing service. Use of an editing service is neither a requirement nor a guarantee of acceptance for publication.

Authors’ response: We wish to express our sincere appreciation to the invaluable inputs given us, we believe we have now strengthened our paper and here we are in a position to reassure you that we have considered language revision and the manuscript has been reviewed meticulously by a native English speaker, whose name is Solomon Girma, who has 11 years of experience as a lecturer of English language and literature in university of Gondar, Ethiopia. Mr Girma is also an assistant instructor of Germen as a foreign language in University of Gondar, Ethiopia. And also necessary corrections made.

  1. In the Introduction part (page 2, line 70-74): Thus, it can serve as an essential pool of

potential blood donors for many eligible adults. In addition, eligible adults, as future adults at community level can take up the role of promoting organ donation by educating and motivating the public to initiate them donates their organs besides their voluntary organ donation.       

 Authors’ response: Many thanks the reviewer for the insightful thoughts and bringing up this issue. Yes the idea is right and we believe that no need of synthesis so we removed it.

  1. c) 4. Need of revision Variables and Measurements part

Authors’ response: We are grateful for providing us a second chance to consider our manuscript, and we also very much appreciate your suggestions, which have been very helpful in improving the manuscript. We also thank the reviewer for their careful reading of our text, after that we tried to amendment all method parts. Please refer it in the revised manuscript.

  1. D) The need of revision the result and discussion parts:

Authors’ response: We greatly appreciate the reviewer’s efforts to carefully review the paper and the valuable suggestions offered. We appreciate the reviewer’s suggestion and we have now corrected line by line in the whole revised manuscript.

Reviewer 2 Report

The manuscript by Zelekes et al describes about the willingness and associated factors affecting blood donation in Gondar town, Ethiopia. Team has done a community based study and found that less than half of the participants were not willing to donate blood. Although 89% of total participant’s the study was aware of blood donation, only 19 % of participant’s out of 547 had had experience in blood donation, therefore it’s important to address the blood donation factor and make people aware about the benefits of blood donation. The community study has covered various aspects such as socio-demographics, knowledge, willingness, blood transfusion related diseases etc.

There are several parts in this article that need further editing and language correction and reference recheck. Overall the materials and methods should be edited and be presented in appropriate format  

Some suggestions/edits are

In page 1 line 8 particularly repeated twice

Page 1 line 45 correct spelling-because

Page 2-Line 49 to line 50- Reference not clear with the information provided in the article

Page 2 line 61- 70, not clear, please rewrite and check for grammatical error

Page 2- line 76 Methods and Methods? – Is it Materials and Methods??

Page 3 under sub title variable and measurements – method not clear please rewrite (line 77 to 117)

Page 3 line 109- please correct font size

Page 2 line 119 – please cross verify the reference

In page 1 and 2- It is written that Total 22 kebeles, but in fig only 21, any reason for that

Page 2   and 3

Fig 1 Please do alignment, would recommend to have full figure in one page, pointers/arrows overlapping

Page 4 line 152- I think it’s better to describe about questioner and details in the beginning of material methods than to the end

Page 5 Table 1, page 6 table 3 and page 7 table 5 - Please realign the table

Page 5 line 196- please correct sentence

Page 5  line 190 page 6 line 204, page 7 line 212, page 8 line 244 , why written as reference not found?

Page 6 line 200 subtitle may be blood donation more appropriate

Page 6 line 216- In that subtitle please unify all numbers written (font size from line 216- 222)

Page 9 Line 285 correct font size

Overall in the paper I would recommend to make it as donate blood instead of blood donate?

Author Response

2022-8-11

Dear; Editor and Reviewers

Please accept our revised manuscript and note our point-by point response to reviewers below for the manuscript titled “Willingness and its factors with blood donation in Gondar Town, Northwest Ethiopia. A community-based cross-sectional study

Our revised manuscript continues to meet the journal’s formal requirements, including the overall word count.

We wish to express our sincere appreciation to the Editorials and the reviewers for their invaluable inputs; we believe we have now strengthened our paper. Please see below a point by point response to the comments raised.

N.B Authors’ responses are indicated in bold italics

Reviewer 2: Some suggestions/edits are

In page 1 line 8 particularly repeated twice

Page 1 line 45 correct spelling-because

Page 2-Line 49 to line 50- Reference not clear with the information provided in the article

Page 2 line 61- 70, not clear, please rewrite and check for grammatical error

Page 2- line 76 Methods and Methods? – Is it Materials and Methods??

Page 3 under sub title variable and measurements – method not clear please rewrite (line 77 to 117)

Page 3 line 109- please correct font size

Page 2 line 119 – please cross verify the reference

In page 1 and 2- It is written that Total 22 kebeles, but in fig only 21, any reason for that

Page 2   and 3

Fig 1 Please do alignment, would recommend to have full figure in one page, pointers/arrows overlapping

Page 4 line 152- I think it’s better to describe about questioner and details in the beginning of material methods than to the end

Page 5 Table 1, page 6 table 3 and page 7 table 5 - Please realign the table

Page 5 line 196- please correct sentence

Page 5  line 190 page 6 line 204, page 7 line 212, page 8 line 244 , why written as reference not found?

Page 6 line 200 subtitle may be blood donation more appropriate

Page 6 line 216- In that subtitle please unify all numbers written (font size from line 216- 222)

Page 9 Line 285 correct font size

Overall in the paper I would recommend to make it as donate blood instead of blood donate?

Authors’ response: Thank you for this positive comment. We appreciate the reviewer’s suggestion and we have now corrected line by line in the whole revised manuscript.

  For example, clarity, font issue (Page 9 Line 285 correct font size), spelling, appropriate words problem(blood donate change to donate blood),table alignment problems was corrected ).

Authors’ response: This concern is noted and well taken. We have now revised the table and added the references. We tried to correct the heading of each table. We appreciate this observation. We have now made the necessary changes and chose to include the information requested by the reviewer in the whole Table.   

Finally, we have corrected the whole manuscript as standard submission guiltiness  

Round 2

Reviewer 2 Report

The manuscript by Zelekes et al describes about the willingness and its factors affecting blood donation in Gondar town, Ethiopia. Team has done a wonderful community based study and found that less than half of the participants were not willing to donate blood and it’s important to address this issue and make people aware of benefits of blood donation. The community study has covered various aspects such as socio-demographics, knowledge, willingness, blood transfusion related diseases etc. After revision the materials and methods has being improved and presented well.

Just a minor suggestion, please check the year in abstract, table and text as Oct 2022 is yet to come. Please correct it to 2021.

Author Response

Reviewer 2: Some suggestions/edits are

In page 1 line 8 particularly repeated twice

Page 1 line 45 correct spelling-because

Page 2-Line 49 to line 50- Reference not clear with the information provided in the article

Page 2 line 61- 70, not clear, please rewrite and check for grammatical error

Page 2- line 76 Methods and Methods? – Is it Materials and Methods??

Page 3 under sub title variable and measurements – method not clear please rewrite (line 77 to 117)

Page 3 line 109- please correct font size

Page 2 line 119 – please cross verify the reference

In page 1 and 2- It is written that Total 22 kebeles, but in fig only 21, any reason for that

Page 2   and 3

Fig 1 Please do alignment, would recommend to have full figure in one page, pointers/arrows overlapping

Page 4 line 152- I think it’s better to describe about questioner and details in the beginning of material methods than to the end

Page 5 Table 1, page 6 table 3 and page 7 table 5 - Please realign the table

Page 5 line 196- please correct sentence

Page 5  line 190 page 6 line 204, page 7 line 212, page 8 line 244 , why written as reference not found?

Page 6 line 200 subtitle may be blood donation more appropriate

Page 6 line 216- In that subtitle please unify all numbers written (font size from line 216- 222)

Page 9 Line 285 correct font size

Overall in the paper I would recommend to make it as donate blood instead of blood donate?

Authors’ response: Thank you for this positive comment. We appreciate the reviewer’s suggestion and we have now corrected line by line in the whole revised manuscript.

  For example, clarity, font issue (Page 9 Line 285 correct font size), spelling, appropriate words problem(blood donate change to donate blood),table alignment problems was corrected ),year of study as you comment ,we corrected it.

Authors’ response: This concern is noted and well taken. We have now revised the table and added the references. We tried to correct the heading of each table. We appreciate this observation. We have now made the necessary changes and chose to include the information requested by the reviewer in the whole Tables.   

Finally, we have corrected the whole manuscripts as standard submission guiltiness